# Implementation of a Delayed Prescribing Model to Reduce Antibiotic Prescribing for Suspected Upper Respiratory Tract Infections in a Hospital Outpatient Department, Ghana

**DOI:** 10.3390/antibiotics9110773

**Published:** 2020-11-04

**Authors:** Sam Ghebrehewet, Wendi Shepherd, Edwin Panford-Quainoo, Saran Shantikumar, Valerie Decraene, Rajesh Rajendran, Menaal Kaushal, Afua Akuffo, Dinah Ayerh, George Amofah

**Affiliations:** 1Public Health England North West Health Protection Team, Liverpool L3 1JR, UK; wendi.shepherd@phe.gov.uk; 2Liverpool School of Tropical Medicine, Liverpool L3 5QA, UK; 248964@lstmed.ac.uk; 3Warwick Medical School, University of Warwick, Coventry CV4 7HL, UK; Saran.Shantikumar@warwick.ac.uk; 4Public Health England National Infection Service, Liverpool L3 1JR, UK; Valerie.decraene@phe.gov.uk; 5Mid Cheshire NHS Foundation Trust, Crewe CW1 4QJ, UK; rajeshrajendran@nhs.net; 6LEKMA Hospital, Accra, Ghana; menaal.kaushal@gmail.com (M.K.); afuaakuffo@yahoo.com (A.A.); dinash51@yahoo.com (D.A.); george.amofah@ghsmail.org (G.A.)

**Keywords:** antimicrobial resistance (AMR), antimicrobial stewardship (AMS), delayed/back-up prescribing, upper respiratory tract infections, developing countries, LMICs, Ghana

## Abstract

*Background*: High levels of antimicrobial resistance (AMR) in Ghana require the exploration of new approaches to optimise antimicrobial prescribing. This study aims to establish the feasibility of implementation of different delayed/back-up prescribing models on antimicrobial prescribing for upper respiratory tract infections (URTIs). *Methods*: This study was part of a quality improvement project at LEKMA Hospital, Ghana, (Dec 2019–Feb 2020). Patients meeting inclusion criteria were assigned to one of four groups (Group 0: No prescription given; Group 1; Patient received post-dated antibiotic prescription; Group 2: Offer of a rapid reassessment of patient by a nurse practitioner after 3 days; and Group 3: Post-dated prescription forwarded to hospital pharmacy). Patients were contacted 10 days afterwards to ascertain wellbeing and actions taken, and patients were asked rate the service on a Likert scale. Post-study informal discussions were conducted with hospital staff. *Results:* In total, 142 patients met inclusion criteria. Groups 0, 1, 2 and 3 had 61, 16, 44 and 21 patients, respectively. Common diagnosis was sore throat (73%). Only one patient took antibiotics after 3 days. Nearly all (141/142) patients were successfully contacted on day 10, and of these, 102 (72%) rated their experiences as good or very good. Informal discussions with staff revealed improved knowledge of AMR. *Conclusions:* Delayed/back-up prescribing can reduce antibiotic consumption amongst outpatient department patients with suspected URTIs. Delayed/back-up prescribing can be implemented safely in low and middle-income countries (LMICs).

## 1. Introduction

Resistance to antimicrobials poses a substantial threat to individual and public health. Antimicrobial resistance (AMR) is responsible for around 700,000 deaths globally per annum—this figure is predicted to rise to 10 million by 2050 if current trends continue unabated [1] with a disproportionately heavier burden in developing countries [2]. Antimicrobial stewardship (AMS) as an organisational, healthcare system-wide approach to promoting and monitoring judicious use of antimicrobials to preserve their future effectiveness has a critical role in reversing these trends [3].

Upper respiratory tract infections (URTIs) antibiotic prescriptions account for the vast majority of antibiotic prescribing—usually in primary care [4]—where they are frequently prescribed for conditions where there is limited evidence of benefit, including acute otitis media and pharyngitis, and where there is no evidence of benefit, such as the common cold [5,6,7]. Delayed/back-up prescribing (where antibiotics can be accessed at a later time after the initial consultation) [8] is one strategy that can be implemented to reduce antibiotic prescribing.

Current British National Institute for Clinical Excellence (NICE) guidance suggests that a delayed antibiotic prescribing strategy “encourages self-management…but allows a person to access antimicrobials without another appointment if their condition gets worse” [9]. Delayed/back-up prescribing should not be used where there is evidence of serious illness or complications, or where the patient is in a clinical risk group [10].

A Cochrane systematic review identified 10 randomised clinical trials that investigated the effectiveness of delayed and no prescriptions strategies for respiratory tract infections [11]. The review found that there was no difference for adverse effects or results favoured delayed antibiotics over immediate antibiotic prescribing; significant reduction in antibiotic use compared to immediate prescription; patient satisfaction favoured delayed prescribing over no antibiotics and there was no difference in patient satisfaction.

Delayed/back-up prescribing can be implemented in a wide variety of ways. An English Ipsos-MORI survey of 1625 participants in 2015 showed that 15% of participants that were prescribed an antibiotics received a delayed prescription [12]. The same study showed a lack of awareness by the public of what the term “delayed prescription” means—after explanation, just 30% of respondents were opposed to General Practitioners (GPs) using this prescribing method for throat, urinary tract, ear, or chest infections. Furthermore, another study [13] reported that delayed prescribing is acceptable no matter how the delay is operationalised, but explanation of the rationale is needed and care taken to minimise mixed messages about the severity of illnesses and causation by viruses or bacteria. A Randomised Control Trial (RCT) considering delayed antibiotic prescribing for uncomplicated acute respiratory tract infections surmised that the practice of delayed prescribing was “associated with slightly greater but clinically similar symptom burden and duration and also with substantially reduced antibiotic use when compared with an immediate strategy” [14].

Delayed/back-up prescribing can be undertaken or approached in different ways. For example, practitioner-centred (when the health professional is responsible for completing the delayed prescription process) or patient-centred, where the patient has responsibility. An advantage of delayed/back-up prescribing is that it provides clinicians and patients with a safety net should an infection deteriorate or fail to improve. Other options may include a systems approach (if diagnosis is clearly identified) whereby local prescribing only allows for delayed prescription—however this may have significant limitations as it does not include the opportunity to review the patient’s condition and consider appropriateness of delayed prescribing. Different approaches may be more appropriate for separate patient groups, practitioners, health facilities, or health systems. A typography of approaches is shown in Appendix A.

Ghana has high levels of AMR, with one study showing multidrug resistance rates of over 75% for some organisms [15]. This demonstrates an urgent need to introduce models of care that optimise antibiotic prescribing within a Ghanaian setting [16].

In 2019, a health partnership between the UK Faculty of Public Health (Africa Special Interest Group) and Ghana Public Health Association secured a global volunteering grant from the Fleming Fund’s Commonwealth Partnerships for Antimicrobial Stewardship (CwPAMS), supported by Tropical Health and Education Trust (THET) and Commonwealth Pharmacists Association (CPA) to undertake a series of stewardship programmes at LEKMA hospital in Ghana. It was felt that there was an opportunity to try different models of antibiotic prescribing within a Ghana healthcare setting to understand what the barriers would be for implementation of a change to existing antibiotic prescribing practices in a low to middle-income country (LMIC) context. Theoretical work to understand challenges to tackling AMR in LMICs has demonstrated that a broad range of factors such as weak governance and poor regulatory measures, compounded by low public awareness of AMR and technological limitations to adequate surveillance, may present complexities not present in non-LMIC settings [2].

The main aim of this study was to explore the feasibility and practical application of different delayed/back-up prescribing models of antibiotics for the management of URTI within a large outpatient facility in a LMIC. This study also aimed to determine if delayed prescribing was safe within this setting, and to test the model’s acceptability to both patients and clinicians.

## 2. Results

### 2.1. Quantitative Results

Over a 3-month period from December 2019 to February 2020, 142 patients who attended LEKMA hospital outpatient’s department and were cared for by one of three medical doctors were eligible for delayed/back-up prescribing. Of these, 86 (61%) were female, 53 (37%) were male; and 3 (2%) did not specify gender (Table 1).

With regard to the different models of delayed/back-up prescribing, 61 (43%) patients were managed conservatively without a back-up prescription or reassessment option (Group 0), 16 (11%) had a post-dated prescription issued (Group 1), 44 (31%) were offered a follow-up appointment for reassessment with a nurse in 3-days if required (Group 2), and 21 (15%) had a prescription left for collection at the hospital pharmacy (Group 3) (Table 1).

As shown in Table 2, nearly half the participants (67, 47%) were working age adults, followed by children under 10 years of age (52, 36%).

The most common clinical diagnoses were sore throat (72%; n = 102), common cold (15%; n = 22) and acute sinusitis (5%; n = 10) with a similar distribution across the four groups (Table 3). Clinical diagnosis was not recorded for three participants. All participants were successfully contacted at day 10 to record outcome data. In all, only 12 (9%) patients remained mildly symptomatic at day 10, although they all indicated they were feeling better and none had sought further healthcare advice. A lower proportion of those in Group 0 had symptoms at day 10, compared with the other groups. Only one individual in the entire eligible patient population (from Group 3) took antibiotics based on worsening symptoms after 3 days. This patient was diagnosed with sore throat, which had subsided when contacted at day 10 and they had completed course of antibiotics.

A Likert Scale of 1–5 was used which has been found to have the most reliability and validity of available Likert Scale methodologies [17]. It should be noted that gaps in the dataset exist where patients were contacted for follow-up but either did not answer questions or the answers were not recorded. Most patients [102 (72%)] rated their experience as good or very good. No patients rated the care they received as poor. When considered by group, the group not given a back-up prescription or an appointment for reassessment (Group 0) were most satisfied with their experience (Table 3).

No adverse events or serious deterioration of illness were reported as a result of the delayed/back-up prescribing model during the 10-day follow-up period for all participants.

### 2.2. Informal Discussions

Following the period of data collection, informal/unstructured discussions were held with LEKMA Hospital staff (doctors, nurse practitioners and pharmacists) involved in the project. This approach, rather than structured interviews, was adopted to give the LEKMA healthcare workers the flexibility to discuss the key components of the pilot from their perspective and important considerations for their patients. These conversations were based on the rapport that had been built between the researchers and the staff at the hospital during the study. This approach, however, meant that each discussion was unique. From these discussions, we were able to ascertain both areas for development as well as successes to inform future delayed/back-up prescribing projects in LMICs.

The main areas for consideration all appear to stem from the lack of visible senior clinical leadership from project implementation. The senior team at LEKMA Hospital were all very supportive of the project from the outset, but the promotion of the project in the outpatients’ department was delegated to the project/study staff. This resulted in misunderstanding by staff about the project aims and the role of clinical staff in delivery of the project. Inevitably, this comprehension impacted on the recruitment of participants to the project as clinical staff did not wish to engage. These issues were addressed once dedicated training had been delivered to the outpatients’ department clinical staff, together with the active promotion of the project by senior leadership.

In-depth understanding of some of the root-causes for AMR and the potential solutions are not commonly/widely shared in Ghana—either by clinicians or by patients. The project provided an opportunity to raise awareness and educate both groups on the importance of the issue via face-to-face discussion with patients and regular AMR training sessions for staff in the outpatients’ department. Furthermore, dedicated nurse practitioners input to the project provided opportunities for nurse practitioners to discuss other aspects of health and healthy living with the patients—this was particularly important as there is a low level of literacy in Ghana [16], and therefore traditional health messages via printed media may be missed.

## 3. Discussion

Our project setting was LEKMA, Greater Accra (Ghana), which has a population of 263,631 representing approximately 5.7% and 0.92% of the Greater Accra and Ghanaian populations, respectively. The population is young, 52% female, with a broad-based population pyramid which tapers towards the top, with very few individuals aged over 65. The catchment area population for LEKMA district general hospital is like that of most hospitals in LMICs, especially in Africa. Therefore, considering some of the cultural and socioeconomic factors, we believe the findings of this study are likely to be applicable to most LMICs, especially African countries.

There was an initial cost to this project (nurse practitioners’ time) which, if the model was to be implemented on a long-term basis, would need investment. Depending on the funding arrangements within the setting of application, this could be viewed as a long-term investment initiative through the reduction of antibiotic prescribing and associated reduction in AMR which will offset the on-going staff support costs. The provision of nursing support did not affect the patient experience or clinical outcome—rather this reassured medical colleagues about the safety of the model—so delayed/back-up prescribing within an LMIC setting for URTIs could be a cost-neutral endeavour if medical support for the model is in place from the outset. More detailed work on the financial aspects of delayed prescribing in LMICs is required to determine the specific cost-benefit of the strategy, but this is likely to be dependent on the specific funding structures for healthcare in local settings.

The primary message from this project is the practical demonstration and evidence that delayed prescribing models for URTIs can be safely utilised in primary care in the Ghanaian healthcare setting—albeit with some areas which require further redress for sustainability and professional acceptance.

### 3.1. Safety and Patient Outcomes

This study demonstrates that all the models of delayed prescribing, including the group given no further follow-up options (Group 0), were acceptable to both staff and patients. With the exception of one patient, all other patients involved in the project reported no deterioration of symptoms as a result of participation. Indeed, there were no adverse events related to the patients’ presenting symptoms during the project and follow-up period (10 days post presentation).

Most patients (91%) who participated in the study had indicated presenting symptoms had resolved by day 10. This reinforces the findings and conclusions from several studies which have demonstrated no differences in antibiotic prescription rates or clinical outcomes between immediate and delayed prescription and, in the short term, there is also little difference in symptom control between delayed prescription, no prescription, or immediate prescription. Delayed/back-up antibiotic prescription resulted in the minority of patients using antibiotics, and any strategy of delayed prescribing is likely to result in fewer than 40% of patients using antibiotics [17]. In this project, a significantly lower proportion of those in Group 0 (2%) had symptoms, albeit milder, at day 10 compared with the other groups. There is a possibility that these patients were generally less unwell at time of presentation, so clinicians felt more comfortable in not providing any treatment or delayed/back-up prescribing options.

It has been found that patients who had been prescribed an antibiotic for cough in the previous two years were over twice as likely to consult for a similar illness, and that a delayed antibiotic prescription strategy reduced re-consultation by 78% in this group [11]. Another study into paediatric antibiotic use also found that “delayed (rather than immediate) antibiotics reduced re-consultations for deterioration for children with URTI in RCT” [18]—a view supported by other studies into antimicrobial prescribing strategies [18,19]. Although the follow-up period in this project was short (10 days), none of the participants sought further heath advice or took antibiotics from any other source. There was also no need for re-consultation in the follow-up period. A longer period of follow-up, however, is required to confirm that there was no further deterioration or need for clinical re-assessment after the 10-day period, as well as the long-term impact on repeat antibiotic usage in the project population.

The research team had expected that there may be some logistical barriers to uptake of different models within the project, such as travel time and expense to return if symptoms did not spontaneously resolve or if clinical condition deteriorated, which may result in poorer clinical outcomes. It is acknowledged that we did not have a complete response to these questions from all participants (39/82 in Groups 1–3), but 37/38 (97.4%) of those who did respond stated that these factors were not an issue meaning that these factors were not a primary concern for patients in the LEKMA project. Although care should be taken in interpreting these results as they are based on a small number of responses, it is not unreasonable to suggest these results are likely to generalisable to the wider catchment area of LEKMA Hospital and beyond in Ghana, i.e., given the similar geographic or population economic circumstances.

### 3.2. Reduction in Antibiotic Prescribing

Over the course of the project, there was a reduction of at least 141 antibiotic prescriptions (one prescription per individual but some individuals may have received multiple prescriptions to treat the same infection). It is reasonable to extrapolate the potential number of antibiotic prescriptions that could be saved over a year if this service improvement project is extended to all clinical staff. Based on surveillance data from LEKMA Hospital, and assuming that all patients presenting with URTI will be prescribed antibiotics [personal communication with LEKMA hospital doctors] we estimated that at least, well over 2000 antibiotic prescriptions can be avoided for just two of the commonest URTIs in LEKMA hospital (common cold and sore throat) in the peak URTI season (October to December), i.e., if all LEKMA hospital outpatient clinicians were involved in the implementation of delayed/back-up prescribing. This figure could be at least 3× higher (>6000 antibiotic prescriptions) if delayed/back-up prescribing is implemented for a full 12-month period for all URTI and by all out-patient department clinicians (Figure 1).

### 3.3. Behaviour Change and Acceptability

The delayed prescribing model was rated as good or very good by 95% of patients. After initial trepidation about the model by clinicians, staff awareness sessions and visible senior clinical leadership were successful methods of ensuring clinician buy-in to the delayed/back-up prescribing model.

Contrary to the perceptions of some local clinicians, Group 0 were the most satisfied with their experience. LEKMA clinicians felt that a potential barrier to the project was that individuals would not wish to leave the outpatient department without a prescription of some nature. This mirrors research considering the issue of increasing antimicrobial resistance, where the authors concluded that “where clinicians feel it is safe not to prescribe antibiotics immediately for people with respiratory infections, no antibiotics with advice to return if symptoms do not resolve is likely to result in the least antibiotic use while maintaining similar patient satisfaction and clinical outcomes to delaying prescription of antibiotics. Where clinicians are not confident in using a no antibiotic strategy, a delayed antibiotics strategy may be an acceptable compromise in place of immediate prescribing to significantly reduce unnecessary antibiotic use for URTIs, and thereby reduce antibiotic resistance, while maintaining patient safety and satisfaction levels” [1].

Our findings demonstrate that, not only was the use of no prescribing (delayed/back-up) acceptable to patients, it was also safe. In addition, the delayed/back-up strategy meant that medical staff were reassured that patients had a point of contact in the nursing team if their condition did not improve/deteriorated, so that they could access further advice and treatment—even if they had been assigned to the Group 0. We hope that these findings can contribute to changing the behaviour of healthcare professionals when considering whether or not to prescribe antibiotics for URTIs.

The experiences of staff during the project, while positive overall, indicate the complexities of the healthcare system in Ghana, the need for complete transparency as to the rationale for such a programme, and the importance of visible senior leadership from the outset. The need for dedicated training and awareness raising of the scheme and the rationale prior to commencement of the project, in addition to senior clinical leadership, cannot be overemphasised.

### 3.4. Strengths of Project

This project contributes to the evidence base around the use of delayed prescribing as a strategy to reduce antibiotic usage within outpatient settings in LMICs. We have found no previous studies that have examined or implemented the strategy of delayed/back-up prescribing in LMICs in our extensive literature searches on Medline, CINAHL, or Global Health databases until July 2020.

The real-life setting of this project provides evidence for the applicability of delayed/back-up prescribing models in similar settings. Through implementation of the project in a functioning and very busy outpatient department, we were able to confirm theoretical principles into practice. Furthermore, the use of different grouping to test all the current suggested delayed prescribing models, including an information only option (Group 0), demonstrates that there was no major difference in outcomes based on the model used.

The use of Group 0 also removes doubt regarding the possibility that delayed/back-up prescriptions may have been used on those who did not require them. Coupled with proactive efforts to reduce private antibiotic sales from local community pharmacies, we can be confident that the results outlined above are an accurate reflection of the treatment each participant received for their condition.

Use of a 10-day follow-up period within the project increased patient safety as it ensured that all patients had a point of rapid access in case of lack of improvement in their condition. Provision of a call to the patient at the end of the follow-up period allowed documentation of clinical outcome.

Finally, the project had an excellent response rate from participants—both on enrolment and at the end of the follow-up period—which provides assurance as to the accuracy of the findings.

### 3.5. Limitations of Project

This service improvement project had some limitations which require discussion to build on our experiences for future delayed/back-up prescribing projects in similar settings.

This initiative had a relatively small number of eligible participants. In part, this may be due to buy-in and understanding of the project from clinicians within LEKMA outpatients’ department in the early period of the project. Although training was offered, it did not reach all outpatients’ department doctors, and this may have contributed to the lack of engagement from all the doctors who worked at LEKMA outpatients’ department at the time. We did not interview clinicians who were not involved in the project to see what factors may have been responsible for the low take-up of the model by prescribing staff.

Although we have one year’s data, we do not know the true trend of URTIs at LEKMA Hospital. Further analysis on general rates of URTIs over the project period may also yield information about whether there was simply a lower burden of URTIs than expected compared with previous years. It was not possible to run the project over all outpatient clinics throughout the week due to project resources only being available on weekdays, so we do not have information on weekend and out of hours attendance for URTIs. We did not collect information on patient’s medical history or existing comorbidities, and these may have impacted on patient outcomes, i.e., persistence of illness or need for antibiotics. In addition, outcomes were self-reported and not validated by clinical examination.

More extensive qualitative and quantitative research throughout the project would have been beneficial to understand other reasons for low take-up—for example, we do not have information on number of potential participants approached who did not consent to being in the project, the number of patients with an URTI who came to the clinic who were not included in the study who may have been given antibiotic treatment. We also do not have information on the exact impact of COVID-19 on potential participant attendance at the outpatients’ department, especially in February 2020. These low numbers of participants, and the single site setting, may limit the generalisability of these findings and inference to other settings in Ghana and other LMICs. This is compounded by this not being an RCT design, and therefore a comparative, standardised study design was not undertaken.

Data quality and data completeness have affected the comprehensiveness of the findings, as there are significant gaps in participant responses—particularly towards the end of the project period when staff were undertaking additional duties to assist with the COVID-19 efforts. Counter to this, at the start of the project, participants were not sequentially allocated to a group, which may have led to some bias. Staff involved in the project have commented that they were not aware of the importance of sequential allocation initially. It was also not possible to blind the study due to the service nature of the project, which may have affected our outcomes. Additional training of project staff in service improvement methodology prior to the commencement of the project may have increased local ownership and understanding of the rationale behind the data collection tools which may have improved data quality. This would have the added benefit of allowing local adaptation of tools to fit local circumstance based on local knowledge.

We recommend that further studies are conducted to address some of these issues in other settings.

## 4. Materials and Methods

The service improvement pilot ran from December 2019 to February 2020 within the outpatients’ department of LEKMA Hospital, Accra, Ghana. Inclusion criteria were all patients who presented at the setting with URTI symptoms during the project period and were deemed to be eligible for the study by the examining clinician. Exclusion criteria were any patient who is diagnosed with URTI but clinician considered that delayed antibiotic prescription was inappropriate or any patient that did not verbally consent to take part. As this was not clinical research, the quality improvement project was discussed with LEKMA Hospital management, and all available evidence regarding back-up/delayed prescribing was presented. The consensus by LEKMA management was that this is evidence-based good practice that would be a key component of quality improvement for the hospital and would not alter patient choices and opportunities; therefore, no ethics approval was required.

There were four different models of delayed/back-up prescribing—no prescription given (only information leaflet); post-dated prescriptions being given to patients at time of first clinical appointment to use if no symptom resolution 3 days after clinic visit; rapid access to a nurse-led clinic for re-assessment after three days, i.e., if symptoms did not improve/patient’s condition deteriorate; and prescription forwarded to the hospital pharmacy and clinician/nurse practitioner asking patient to visit the hospital pharmacy to collect a pre-written antibiotic prescription if symptoms did not improve within 3 days. Delayed/back-up antibiotic prescribing was only to take place when it was deemed clinically appropriate to do so—this decision was at the sole discretion of the clinician responsible for the patients’ care. The clinical pathways are available in Appendix A.

Two experienced nurses were recruited from the hospital staff to provide support to the project on a full-time basis for the project duration, although their time was diverted to COVID-19 response towards the end of the project. A data collection tool was developed using Microsoft Excel to ensure contemporaneous data capture by these nurses and provide a database of clinical presentation, clinical outcome at the end of follow-up period, and general evaluation information. Patient information leaflets were prepared for different groups. All project materials are available in the Appendix A.

Participants were allocated to one of the four groups on a sequential basis by the nursing team. The study was designed so that there was sequential randomisation of participants but, in reality, this was initially more ad-hoc than was planned which resulted in unequal sized groups between the different interventions and may have introduced bias into the study.

Dedicated face-to-face training to medical, nursing, and administrative staff in the outpatients’ department and pharmacy was provided by the UK partnership staff in advance of the project to advise them of the project aims and objectives. Further training was provided in January 2020 by senior medical staff at LEKMA Hospital.

Community pharmacists in the area surrounding LEKMA Hospital were visited by the project team to try to minimise non-prescribed antibiotic purchases direct from pharmacists.

All patients were contacted via telephone 10-days after their initial presentation to ascertain if they were still symptomatic; if they had consulted other medical professionals over the 10-day period; antibiotics taken in previous 10-days (and source); and their experience of the care they received using a Likert scale.

Informal discussions with LEKMA staff members actively involved in the project were undertaken, and their transcripts were analysed to understand how the field experience at the local level in Ghana related to the potential barriers outlined in other work on antimicrobial prescribing in LMICs [2].

## 5. Conclusions

Despite good evidence for a delayed prescribing approach in other regions (mainly high-income countries), there is sparse published evidence for the use of delayed/back-up prescribing in LMICs—particularly in Africa, where alternative methods of gaining antibiotics may exist (such as direct illicit purchase from community pharmacists, as can occur in Ghana). Furthermore, in LMIC hospital outpatient departments, which see new clinical presentations of illness (much like in primary care), the proportion of antibiotic prescriptions that are written for URTIs is unknown, and inappropriate antibiotic prescribing is likely to be high.

The results from this service improvement project show support from both clinicians and patients for more dedicated interventions to reduce inappropriate prescribing of antibiotics in LMICs with little preference for which model of delay/back-up prescribing used. The success of the models is reflected through a significant reduction in antibiotic use for URTIs in LEKMA outpatients during the project with no serious illness or adverse events recorded over the 10-day follow-up period. Furthermore, upscaling implementation delayed/back-up prescribing in LMICs could contribute to improvement in clinicians’ confidence, optimise antibiotic prescribing and reduce antimicrobial resistance.

Further in-depth exploration of clinicians’ and patients’ experiences and perceptions need to be captured to help optimise delayed/back-up prescribing implementation. Extended project schemes along the same model should be used in different settings and with larger cohorts of patients to prove the clinical applicability of the model to other LMIC settings using a bigger data set.

## Figures and Tables

**Figure 1 antibiotics-09-00773-f001:**
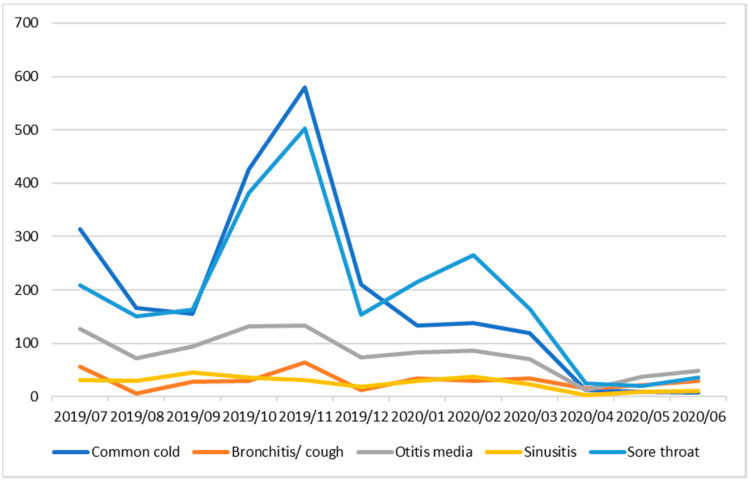
Attendance of Patients with upper respiratory tract infections (URTIs) at LEKMA Hospital, July 2019–June 2020.

**Table 1 antibiotics-09-00773-t001:** Group description and characteristics of participants.

Characteristics			Group			
			0	1	2	3
	Description		Not considered in need of a back-up prescription. However, they were given a leaflet that outlines the reasons for the clinical decision	Post-dated prescription (given to patient)	Rapid reassessment by nurse at 3-days post initial presentation	Post-date prescription forwarded to hospital pharmacy
**Participants** **(n = 142)**	(% of total)		61 (43%)	16 (11%)	44 (31%)	21 (15%)
**Sex**	(n, % of group)	Females	40 (66%)	9 (56%)	24 (55%)	13 (62%)
**Age distribution**	(median, IQR in years)		22.5 (2–49)	19 (4–63)	17 (4–30.5)	24 (5–46)

**Table 2 antibiotics-09-00773-t002:** Age Profile of Participants.

Age Band	Number (n = 142)	%	Cumulative Age Band	%
0–10 years	52	6%	0–10 years	37%
1–10 years	43	30%		
11–19 years	11	8%	11–65 years	47%
20–45 years	37	26%		
46–65 years	19	13%		
≥66 years	13	9%	≥66 years	9%
Not recorded	10	7%	Not recorded	7%

**Table 3 antibiotics-09-00773-t003:** Characteristics and outcomes of participants by group where recorded.

Characteristics			Group			
			0	1	2	3
**Diagnosis**	(n, % of group)	Sore throat	46 (75%)	10 (63%)	33 (75%)	13 (62%)
		Common cold	9 (15%)	3 (19%)	7 (16%)	3 (14%)
		Sinusitis	3 (5%)	1 (6%)	4 (9%)	2 (10%)
		Other	2 (3%)	1 (6%)	0 (0%)	2 (10%)
		Not specified	1 (2%)	1 (6%)	0 (0%)	1 (5%)
**Symptoms at day 10**	(n, % of group)		1 (2%)	4 (25%)	3 (7%)	4 (19%)
**Antibiotics taken**	(n, % of group)		0 (0%)	0 (0%)	0 (0%)	1 (5%)
**Experience reported as good/very good**	(proportion of respondents, %)		57/57 (100%)	5/5 (100%)	30/34 (88%)	10/11 (91%)

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
