# Peer review of "Implementation of a Delayed Prescribing Model to Reduce Antibiotic Prescribing for Suspected Upper Respiratory Tract Infections in a Hospital Outpatient Department, Ghana"

_antibiotics, 2020, doi:10.3390/antibiotics9110773_

Round 1

Reviewer 1 Report

The authors discuss and attempt to tackle the important problem of over-prescription/use of antibiotics and the development of antimicrobial resistance in low-middle income countries. Although this is work is vital, there are flaws in the methodology that make it inappropriate for publication at this time: 

  • The methods section needs extensive revision and there are several important questions that need to be answered: 
    • How was randomization into groups achieved? Sequential assignment after eligibility?
    • What are the specific inclusion criteria? 
    • What are the specific exclusion criteria? 
    • How was patient eligibility determined?
    • How were the 3 medical doctors selected and why? 
    • Were the nurses and medical doctors blinded to the assignment? 
    • What is the primary endpoint of the study? It is unclear if the primary endpoint is patient-reported symptoms, patient-reported satisfaction, or antibiotic use? And how were the endpoints determined?
    • What are the secondary endpoints of the study? 
    • How were clinical diagnoses determined? i.e. was the principle ICD-9/10 code of the visit used? Or was there a separate report filed specifically for the study?
    • Where patient comorbidities/medical histories collected? If not, that may be an important variable affecting the outcomes. These could be collected using ICD-9/10 codes if available
    • Further discussion/definition of the different prescribing methods and how they were carried out or standardized
    • A bivariate or multivariate analysis describing predictors of the primary and secondary outcomes would be useful as descriptive statistics do not provide a detailed analysis and do not account for confounding factors

The discussion and limitations sections will need to be subsequently revised and re-written after all the above-stated questions/issues are addressed. 

Reviewer 2 Report

This is an interesting contribution to our understanding of how antimicrobial stewardship, the delayed prescribing models, helps to reduce the antibiotic prescriptions for URTI in Ghana, which further provides fresh insights into the improvement of appropriate use of antibiotics in the LMIC context. The study provides a very detailed context, which is easier for readers unfamiliar e.g. Ghana or delayed prescribing models. However, this manuscript needs improvement from these aspects.

  • The general flow of the article should be changed i.e. move section of Materials and Methods forward to Results, to avoid any confusions; also, some sentences could be moved (e.g. line 205-208 should be moved to introduction rather than left in discussion)
  • These is a lack of ethical statement.
  • The qualitative results urgently need more details, such as how many interviews were undertaken, the participant information, a table of themes and subthemes. Also, it would be much more helpful to include some quotes in.
  • Some information discussed in discussion section was not supported by results.
  • Any possible baseline data for comparison? What was previous antibiotic prescription amount/rate for URTI patients in the similar season before the interventions (data from this hospitals or other comparable data)?

Specific comments/concerns

  1. Line 116 nearly half of them (67, 47%) were working age adults. Does it include age groups of ’11-19’, ’20-45’ and ’46-65’ together? Is group ’11-19’ also included in working age adults? Need to be clarify or provide criteria as reference.
  2. Line 127 Most patients [103(95%)]. The total number calculated from the Table 3 is 102 (57+5+30+10). Need to be corrected.
  3. Line 200-202 ‘This figure could be at least 3x…and by all out-patient department clinicians.’ How could 3x (> 6,000 antibiotic prescriptions) obtained? Also, this sentence is a bit confusing – why antibiotic prescriptions will be higher if delayed prescribing is implemented throughout the year? Please clarify or rephrase it
  4. Some Chinese characters ‘月’ appeared in Figure 1? The language should be kept consistent.

Reviewer 3 Report

The manuscript by Chebrehewet and colleagues tackles the problem of antibiotic over-/misuse and antibiotic resistance in low and middle-income countries.  The Authors investigated the effect of the delayed/back-up prescribing strategy used for outpatients with upper respiratory tract infections. The project was carried out in the LEKMA Hospital in Ghana.  Chebrehewet and colleagues demonstrated that the strategy may be effective in reducing antibiotic use. I have only minor comments.

There are some discrepancies between the number of participants presented in Table 1, Table 3 and in the text.  There were 38 participants in Cohort 1 (Table 1), but according to Table 3, only 16 were diagnosed (Sore throat -10, Common cold – 3, Sinusitis -1, Other -1, Not specified-1). There were 23 participants in Cohort 2 (Table 1), but according to Table 3, 30/34 patients from the cohort responded after the 10-day period. Probably, the numbers given in line 187 should also be corrected.

It is not entirely clear how the Authors estimated the reduction in antibiotic prescribing (lines 197-202). For a complete picture, the number of outpatients who were admitted to LECMA but not involved in the project due to antibiotic prescriptions could be given.

The Materials and Methods section, lines 330-332: “All patients were contacted… to ascertain if they had consulted other medical professionals… antibiotics taken in previous 10 days”.  Participants’ responses to these questions should be mentioned in the text.

Reviewer 4 Report

Did the project receive any ethical approval? Please add this information to the methods section.

Please Improve the abstract. The first sentence, with identification of study setting, is methods, not the background. Add the aim of the study.

Description of the methods in the abstract is very confusing. Please rewrite all the abstract.

Lines 59-67 – These transcriptions are not necessary. Please remove it. Write by your own words the main conclusions of the systematic review.

Lines 79-82 – needs a reference.

Line 24 – Please add the full name of LMICS

Did the authors use any topic guide for qualitative analysis? A published one? Or have they constructed one? How was it constructed? Who participated in the interviews? Physicians? Nurses? Pharmacists? Other LEKMA staff members actively involved? Who?

Line 101 – It seems to be missing an "also"

Lines 106-107 – “… patients who attended LEKMA hospital outpatient’s department and cared for by one of three medical doctors were eligible for delayed / back-up prescribing.” What are the inclusion criteria? This information should be added in the methods section.

Lines 116-117 – “As shown in Table 2, nearly half of them (67, 47%) were working-age adults, followed by children 116 under 10 years of age (52, 36%).” This information is not so evident when looking to table 2. Please confirm this information.

Lines 127 and 128 – “ Most patients [103 (95%)] rated their experience as good or very good.” How?

Lines 330-334 All patients were contacted via telephone to give their experience of the care they received using a Likert scale. Did the authors use a validated and published  Likert scale, or did they constructed and validated one? Results of validation are welcome. As well the scale in the supplementary material. Results from the application should be presented. The results reported in the table about experience-reported are very general and it seems that did not reflect the application of a scale. It seems more than the authors asked directly to the patients how did they have considered the experience. Please present more detailed results.

The qualitative results are very general. It is important to improve this section and present the real obtained results from the interviews.

Some care is needed in the conclusion and generalization of the results when representativeness of the population does not exist. How was calculated the sample? Please revise the first sentence of the discussion.

Line 160 –“ … no significant difference in clinical outcomes between the different models used.”  This information on the discussion was not supported by the results because no clinical Outcomes were presented in the results section. Did the authors mean that there are no differences in the infectious symptoms of the patients? During the telephone interview, patients were asked about it or were only asked about the purchase of the antibiotic delayed prescribed?

Lines 186-188 – again information discussed was not presented in the results section.

Please check the results section and the discussion section, and add the results that are missing to support the discussion.

Round 2

Reviewer 1 Report

The authors clearly performed this study to address the very important issue of over-prescription of antibiotics, and I really appreciate their responses to the comments. However, in my initial review, I felt like there were flaws in the methodology that were either due to poor study design or lack of clarity in the text. I still think that the study design and application makes it impossible to make any associative or causal statements. I understand the observational nature of the study, but it seems that the ad-hoc decision making, uneven distribution of patients in to the groups, and issues with standardization in many instances makes publishing the study difficult. Again, I want to reiterate that the work the authors are doing is important, but as is, my initial decision to reject the paper for publication still stands.

Author Response

Comments:

I felt like there were flaws in the methodology that were either due to poor study design or lack of clarity in the text. I still think that the study design and application makes it impossible to make any associative or causal statements. I understand the observational nature of the study, but it seems that the ad-hoc decision making, uneven distribution of patients in to the groups, and issues with standardization in many instances makes publishing the study difficult

Responses:

Thank you for your comments.  We have reworked the paper to remove any indication of causality or association.  We have also changed some of the terminology to clarify that this is a pilot service improvement and not an RCT. 

Reviewer 4 Report

In general, the authors clarified all questions and improved the manuscript.

However, related to the discussion and conclusions I maintain some concerns:

In line 170 authors concluded that “…. No significant difference in clinical outcomes …” I think that clinical outcomes were not evaluated by a doctor. What authors assessed was the patient's perceptions about their symptoms and not clinical outcomes.

The discussion should reflect the results presented on:

- Line 125-126 – “… lower proportion of those in Cohort 0 had symptoms at day 10, compared with the other group…”

- Lines 137-138 - No adverse events or serious deterioration of illness were reported as a result of the delayed/back-up prescribing model during the 10-day follow-up period for all participants.

Please check the following:

Line 291 – 292: “We did collect information on patient’s medical history …” Did Collect or did not collect?

Author Response

Comment 1:

In line 170 authors concluded that “…. No significant difference in clinical outcomes …” I think that clinical outcomes were not evaluated by a doctor. What authors assessed was the patient’s perceptions about their symptoms and not clinical outcomes.

Response 1:

Thank you for highlighting this point.  We have reworded the appropriate section so that it is clearer that any outcomes were self-reported by the patient. 

Comment 2:

- Line 125-126 – “… lower proportion of those in Cohort 0 had symptoms at day 10, compared with the other group…”

Response 2:

Thank you for your comments.  We have clarified this point. 

Comment 3:

- Lines 137-138 – No adverse events or serious deterioration of illness were reported as a result of the delayed/back-up prescribing model during the 10-day follow-up period for all participants.

Response 3:

Thank you for highlighting this point.  We have reworded the appropriate section so that it is clearer that any outcomes were self-reported by the patient. 

Comment 4:

Line 291 – 292: “We did collect information on patient’s medical history …” Did Collect or did not collect?

Response 4:

Thank you for your comments.  We have clarified this point.